# Histological and Transcriptomic Characterization of Full-Thickness Skin Wound Healing in Maraena Whitefish (*Coregonus maraena* Bloch, 1779)

**DOI:** 10.3390/ijms26178315

**Published:** 2025-08-27

**Authors:** Marcin Kuciński, Tomasz Liszewski, Teresa Własow, Anna Wiśniewska, Dorota Fopp-Bayat

**Affiliations:** 1Department of Marine Biology and Biotechnology, Faculty of Oceanography and Geography, University of Gdansk, M. Piłsudskiego 46 Av., 81-378 Gdynia, Poland; 2Department of Ichthyology and Aquaculture, Faculty of Animal Bioengineering, University of Warmia and Mazury in Olsztyn, Oczapowskiego 5, 10-719 Olsztyn, Polandtewlasow@uwm.edu.pl (T.W.); dariama@uwm.edu.pl (A.W.)

**Keywords:** fish, skin wound regeneration, re-epithelialization, inflammation, remodeling, immune response, aquaculture

## Abstract

The healing process of full-thickness skin wounds in maraena whitefish (*Coregonus maraena*) was investigated to provide preliminary insights into the species’ tissue regeneration mechanism and dynamics following mechanical injury-simulating standard aquaculture PIT tagging procedures. A mechanical skin injury was induced on the dorsal flank of one-year-old maraena whitefish using a 15G needle, and skin regeneration was tracked for 15 days post-wounding (dpw). Expression levels of six genes involved in immune response and inflammation (*IL-17D* and *CD-4*), cellular stress response (*HSP-90*), and cell proliferation and tissue growth (*MMP-9*, *p53*, and *TGF-β*) were examined in wounded and intact skin tissues, liver, and head kidney. Histological analyses were also performed to monitor wound-healing progression. Histological examination revealed typical fish wound-healing characteristics involving re-epithelialization on the 1st day post-wounding (dpw), acute inflammation on the 3rd dpw, granulation tissue formation and intensive wound remodeling on the 8th dpw, and full tissue regeneration by the 15th dpw. Gene expression analysis revealed dynamic tissue-specific patterns: *IL-17D* and *CD-4* were upregulated early in wounded skin, indicating rapid immune and inflammation activation, while *MMP-9* and *TGF-β* peaked later, supporting tissue remodeling and regeneration. *HSP-90* and *p53* genes were highly expressed in the mid to late stage of healing, reflecting cellular stress response associated with acute inflammation and a high rate of cell proliferation in wounded skin. Significant transcriptional changes in the liver and head kidney further supported the systemic nature of the wound response and emphasized the importance of immune function in the species’ tissue-repair process. The obtained findings provide novel insights into the mechanisms and dynamics of skin healing in maraena whitefish, potentially supporting the development of improved health management strategies for this species in aquaculture.

## 1. Introduction

In fishes, the skin is a compact, non-keratinized living organ that covers their entire external body surface, including the head, fins, and eyes. It serves as the organism’s primary defense against harmful biological, chemical, and physical factors, while simultaneously preserving water, solutes, and nutrients within the body [1,2,3]. Like in other vertebrates, the skin of fishes exhibits a highly conserved structure, consisting of the epidermis, dermis, and hypodermis [3]. The epidermis, the outermost layer of fish skin, is composed of keratocytes and mucous cells that secrete a continuous, adherent layer of mucus. It contains a diverse range of immune-related substances, such as agglutinins, immunoglobulins, C-reactive proteins, and other antimicrobial peptides [3,4,5]. In addition to protecting against pathogens, the mucus layer also supports functions like respiration, microbiota maintenance, and particle entrapment. In turn, keratocytes act as physical barriers against harmful environmental features. They also exhibit phagocytic activity, containing high levels of peroxidase and other immune factors [3,6,7]. The outer layer of the dermis is composed of loose connective tissue, housing blood vessels, nerve cells, chromatophores, iridophores, and peripheral nerve cells. The inner dermis contains dense connective tissue of tightly packed collagen fibers and specialized structures like scales, which increase the skin’s resistance to puncture. The hypodermis is located beneath the dermal layer and above the muscles. It is mostly occupied by adipocytes, as well as small number of vascular and neural tissues, chromatophores, and leucophores [2,3].

The heightened vulnerability to injuries in aquatic habitats has led fishes to evolve faster and more efficient healing mechanisms compared to terrestrial vertebrates [3,8]. Consequently, wound healing in fishes does not begin with the initial blood-clot formation. Instead, it starts with rapid re-epithelialization by keratocytes migrating from surrounding undamaged tissue, followed by inflammation, granulation tissue formation, and wound remodeling [3,9]. In fishes, the loss of blood and other extracellular fluids after injury is prevented through the vasoconstriction and fibrin aggregation, which forms clot-like structures. Available studies have revealed that this process is accompanied by the upregulation of genes involved in epidermal repair and hemostasis, such as integrins, fibronectin, and genes related to eicosanoid metabolism [3,10]. In general, the rate of wound healing in fishes primarily depends on the species, as well as the type and size of the wound. However, water temperature, stress level, age, and nutritional status are also important factors determining the rate of wound healing in fishes [3,11]. In thermophilic and temperate fish species, re-epithelialization and granulation tissue formation become evident as early as two days post-wounding (dpw). Depending on wound type and size, they can completely regenerate damaged skin within a month. In contrast, re-epithelialization and granulation tissue development in cold-water species typically commence later, often between 10 and 42 dpw, with complete tissue regeneration taking up to 100 dpw [3].

Typically, skin injuries in fish are categorized into mechanical and ulcer-like wounds. Mechanical skin injuries, such as abrasions and cuts, are further classified into (1) superficial wounds, (2) partial-thickness wounds, and (3) full-thickness (deep) wounds [3]. Superficial wounds do not affect the dermis, whereas partial-thickness wounds penetrate both the epidermis and dermis. In turn, deep wounds extend through entire skin layers to the subcutaneous adipose tissue or deeper [3]. In fishes, mechanical skin injuries mainly result from traumatic incidents associated with sudden panic episodes, adverse weather conditions, or encounters with predators [3]. Ulcer-like wounds primarily arise from pathogenic microbial activity and skin lesions linked to various dermatological diseases [3]. Superficial and partial-thickness skin wounds in fishes are known to heal the fastest, often within hours to days, primarily involving re-epithelialization, limited inflammation, and epidermis restoration, along with mucous cell activation [3,5]. In contrast, full-thickness wounds trigger the complete healing cascade, involving a more complex response with regeneration of multiple cell types, tissues, and structures. This includes additional stages, such as scale differentiation, extracellular matrix production, and basal plate matrix formation and its mineralization [3]. Moreover, tissue clefts can hinder keratocyte migration, causing significant delays in deep-wound healing, usually taking weeks to months to heal, depending on wound severity, fish species, and environmental conditions [3,9,12].

In intensive aquaculture systems, fish commonly sustain skin injuries not only due to natural skin disease processes but also because of mechanical trauma incidents that occur under high stocking density conditions [13]. Once the skin barrier is breached, pathogens can easily invade the organism, making the preservation of an intact epidermis vital for maintaining fish health and welfare under aquaculture settings [8,14]. However, the effective implementation of strategies for preventing or treating both immediate and long-term skin damage requires a detailed understanding of the principles of the wound-healing mechanism and its dynamics in farmed fishes. Thus, extensive research has been dedicated over the last decades to investigating skin wound-healing processes in various fish species, with a specific emphasis on understanding the role of skin mucus in humoral defense mechanisms, and factors that may enhance or delay the skin wound-healing process. So far, at least 13 different fish species, including the model one, zebrafish (*Danio rerio*), and commercially farmed species, such as salmonids (Salmonidae), carps (Cyprinidae), breams (Sparidae), and catfishes (Siluriformes), have been studied (reviewed in [3]). Among salmonid fishes, research on skin wounds’ regeneration have primarily focused on Atlantic salmon (*Salmo salar*) and rainbow trout (*Oncorhynchus mykiss*) [11,12,15,16,17,18,19].

The maraena whitefish (*Coregonus maraena*) holds significant ecological and aquacultural importance in Europe. In the Southern Baltic Sea region, the species has a revered status as a cherished traditional culinary delicacy, often served fried or smoked. Due to various human activities, such as water pollution, damming, and overfishing, natural populations of whitefish are considered to be endangered [20]. Consequently, the maraena whitefish is currently being reared under aquaculture both for food production and as stocking material to enhance wild populations through fishery supplementation in open waters [21,22]. However, detailed information on the mechanisms and recovery timeline of skin-injury healing in the maraena whitefish remains scarce, hindering efforts to optimize the species’ commercial aquaculture husbandry protocols.

Therefore, the main objective of this study was to characterize the healing process of full-thickness skin wounds on the dorsal flank of juvenile maraena whitefish specimens over a 15-day period, induced by puncturing with a 15G hypodermic needle. The wounding procedure simulated the standard practice of PIT tagging commonly performed in the aquaculture production of this species. The skin-healing process was studied through histological observations and the analysis of selected gene expressions involved in skin-tissue regeneration.

## 2. Results

### 2.1. Histological Analysis

A full-thickness skin injury on the dorsal body part, just below the dorsal fin, was inflicted in maraena whitefish using a 15G hypodermic needle to histologically characterize the healing process. The mechanical skin injury resulted in the breakdown of epidermis continuity and violation of the dermis, accompanied by bleeding without visible blood-clot formation in all examined fish (Figure 1a–d).

On the first day post-wounding, re-epithelialization of the injury site was nearly complete in all sampled fish. At this stage of wound healing, signs of extensive loss of epidermal cells and hemorrhaging were still visible across the wounded area. In histological images, the wound bed was covered by an amorphous exudate, on which keratocytes were migrating from the wound margins toward the center. The keratocytes observed in the sections displayed a characteristic polygonal shape, centrally located nuclei, and pale-staining cytoplasm. They were also predominantly present in multiple epidermal layers of the undamaged skin adjacent to the wound border in the examined fish. Mucous cells were additionally spotted alongside keratocytes at the migratory front. The wound bed was characterized by the presence of damaged muscle and collagen fibers with no signs of apparent regeneration. Moreover, infiltration of polymorphonuclear inflammatory cells (PMNs), presumably leukocytes, into the wound bed was observed, initiating a strong cellular inflammatory response. Chromatophores (most likely melanocytes), contributing to hyperpigmentation, were observed around the wound margins and remained visible throughout the entire healing process (Figure 1a).

On the 3rd post-wounding day, initial remodeling of the damaged skin site was observed, along with a strong and ongoing cellular inflammatory response in all examined fish. At this healing stage, the wound area was fully covered by a newly formed neo-epidermal layer with an organized appearance; however, it was weakly separated from the wound bed by the membrane. Numerous apically located mucous cells were observed within the formed epidermal layer. Many of them were open to the extracellular space, forming a thick mucus coating over the epithelial surface. Intensive keratocyte proliferation was recorded throughout the entire neo-epidermal layer, although fewer proliferating keratocytes were seen at the wound margins. A thick layer of granulation tissue covering the entire wound bed was recorded. The wound bed was primarily characterized by disorganized tissue fibers, with the presence of proliferating fibroblasts. Cell proliferation was identified based on the presence of densely packed nuclei within the regenerating epidermis and increased cellularity in the underlying tissue. The most intensive cell proliferation was observed near the border between the intact skin. Moreover, vacuolization and muscle tissue degradation, along with presence of polymorphonuclear inflammatory cells (PMNs), presumably leukocytes, were spotted across the wound bed. Enlarged chromatophores (most likely melanocytes) were also observed within the newly formed epidermis (Figure 1b).

On the 8th post-wounding day, injury sites were filled with a thin layer of granulation tissue and exhibited partial contraction in all fish, as confirmed by microscopic evaluation at various magnifications. In the wound bed, inflammation and tissue repair were both prominent, accompanied by numerous proliferating cells filling the regeneration area. The epidermal layer showed enhanced organization compared to the previous wound-healing stage, with a clear separation from the wound bed and increased thickness in the wound center compared to the surrounding regions. At the wound margins, intensively proliferating cells were organized in an oval pattern across all samples. At this stage of skin healing, the wound bed was largely filled with collagen deposits containing quite irregularly oriented fibrils. The fibrotic layer was found to be thicker at the wound center compared to the surrounding regions. Instead of newly formed collagen fibrils, myofibroblast-like cells with elongated phenotypes were also identified in the wound bed. The observed collagen fibril deposits were arranged in rope-like structures extending from the myocommata into the wound bed. Chromatophores (likely melanocytes) were observed infiltrating the fibrotic tissue from deposits beneath the epidermal and collagenous layers, contributing to its additional hyperpigmentation (Figure 1c).

On the 15th day of post-wounding, the wound site exhibited complete contraction and macroscopically became almost indistinguishable from the undamaged skin in all examined fish. Histological views revealed that cell proliferation in the wound bed was markedly less intensive compared to the previous healing stage. All skin layers were also completely regenerated, with the epidermis consisting of a thick layer of keratocytes, together with large mucous cells. The epidermis was also covered by thick layer of mucus. Capillaries were additionally spotted directly beneath the regenerated epidermal layer. Muscle tissue showed nearly complete regeneration alongside elongated myofibroblast-like cells still present in the wound bed. The collagen fibers beneath the epidermis appeared thicker and were running more parallel to the epithelial layer. Chromatophores (likely melanocytes) were observed in two layers of the contracted wound: one just beneath the epidermis and the other below the newly formed collagenous tissue (Figure 1d).

### 2.2. Gene Expression Analysis

Overall, mRNA transcripts of all the investigated genes exhibited distinct organ- and tissue-specific expression patterns, reflecting a systemic response to full-thickness skin injury in the examined maraena whitefish. Among all sampled tissues, a significant upregulation of the analyzed genes was predominantly observed in wounded skin tissue, apart from the *p53 protein* (*p53*) gene, which exhibited the opposite pattern (Figure 2a–f).

A significant (*p* < 0.05) upregulation of *interleukin 17D* (*IL-17D*) gene expression was recorded in all tissues and organs of fish subjected to skin injury between the 1st and 8th day post-wounding (dpw), with the highest increase in mRNA transcription levels detected in wounded skin tissue. The peak expression of the *IL-17D* gene was detected on the 3rd dpw across all examined tissues, with comparable levels except from the intact skin, where expression was significantly lower (*p* < 0.05). By the 15th dpw, *IL-17D* expression was downregulated to levels comparable to the control group across all tissues (Figure 2a).

*Cluster of differentiation 4* (*CD-4*) gene expression showed a significant (*p* < 0.05) upregulation in wounded skin tissue (on the 1st, 3rd, and 15th dpw) and head kidney (on the 3rd and 8th dpw). In turn, a significant (*p* < 0.05) downregulation of *CD-4* expression in the wounded skin and upregulation in the head kidney were recorded on the 8th dpw. Compared to the control group, a significant (*p* < 0.05) downregulation of the *CD-4* expression was also noted in the intact skin tissue of the wounded fish, showing a progressive decrease across all examined time points post-wounding. No significant differences (*p* > 0.05) in *CD-4* mRNA expression levels were detected in the liver (Figure 2b).

A significant (*p* < 0.05) upregulation of *heat shock protein 90* (*HSP-90*) gene expression was detected in both wounded and intact skin tissue between the 1st and 8th dpw, peaking on the 3rd dpw, followed by a subsequent downregulation on the 15th dpw to the levels observed in the control group. The mRNA expression levels of *HSP-90* were significantly (*p* < 0.05) higher in wounded skin tissue compared to intact skin. In the liver of wounded fish, a consistent suppression of the *HSP-90* gene was detected until the 3rd dpw, followed by upregulation on the 15th dpw, reaching levels comparable to those observed in the control group. No significant differences (*p* > 0.05) in *HSP-90* mRNA expression were observed in the spleen (Figure 2c).

A steady but significant (*p* < 0.05) increase in mRNA transcription of the *metalloproteinase-9* (*MMP-9*) gene was recorded in all examined tissues across each time point post-wounding, except from the liver, where differences were insignificant (*p* > 0.05). Peak expression of the *MMP-9* gene across all tissues was observed on the 15th dpw, with the highest transcription level recorded in the wounded skin tissue (Figure 2d).

In the case of *p53 protein* gene (*p53*), a significant (*p* < 0.05) downregulation of mRNA transcription levels was observed from the 1st to the 8th dpw in both wounded and intact skin tissues, reaching the lowest level on the 3rd dpw. In contrast, a significant (*p* < 0.05) upregulation of the *p53* gene expression was recorded in the liver and spleen between the 1st and 8th dpw, peaking on the 3rd dpw. Overall, *p53* mRNA transcription across all examined tissues returned to levels comparable to the control group by the 15th dpw (Figure 2e).

No significant (*p* > 0.05) changes in the expression level of *the transforming growth factor-β* (*TGF-β*) gene were observed in any of the examined tissues, except for the wounded skin, which showed a significant (*p* < 0.05) upregulation on the 15th dpw, reaching the highest expression level among all tissues and organs analyzed (Figure 2f).

## 3. Discussion

In the present study, the healing process of a full-thickness skin wound in maraena whitefish was investigated by means of histological and gene expression analyses. The results showed that complete regeneration following puncture with a 15G hypodermic needle required at least 15 days post-injury. The skin-healing process in the examined species comprised four main stages: (1) re-epithelialization, occurring up to approximately the 1st day post-wounding (dpw); (2) inflammation between the 1st and 8th dpw; (3) granulation tissue formation between about the 3rd to 8th dpw; and (4) wound remodeling, occurring between the 3rd and at least the 15th dpw. Moreover, the recorded wound-healing process lacked a prolonged late remodeling phase and exhibited substantial overlap between the early (inflammation) and late (remodeling) phases of healing. In contrast, studies on Atlantic salmon and rainbow trout using a 5–6 mm punch biopsy to create full-thickness wounds have reported a clear distinction between early (1–14 dpw) and late (36–100 dpw) regeneration phases [12,18]. The results recorded in the present study also revealed that the overall wound-healing dynamics in maraena whitefish are intermediate between those observed in thermophilic and cold-adapted species [3]. However, the observed wound-healing timeline in this species might be attributed to the nature of the inflicted wounds, as the technique used did not cause severe tissue loss.

In fishes, re-epithelialization is a crucial stage of tissue regeneration, as it establishes a protective barrier over the wound bed against pathogens and other harmful external factors [3]. In the present study, intensive re-epithelialization was observed within the 1st dpw in the examined maraena whitefish, mirroring responses previously reported in Atlantic salmon and rainbow trout [12,18]. During re-epithelization in fishes, keratocyte cells migrate collectively as a sheet toward the wound area from all sides to cover the injury. The migration stops when the advancing cell fronts meet each other [3]. It is considered that the primary source of recruited keratocytes comes from the inter-scale pockets, highlighting their significance in wound healing in fishes [9]. During re-epithelization, keratocytes undergo structural changes, with superficial keratocytes flattening and developing microridges that enhance the epidermal surface for effective mucus retention, as well as gaseous and ionic exchange [23]. Similar to Atlantic salmon and rainbow trout, the examined maraena whitefish developed an amorphous exudate on the wound surface during regeneration that helps early keratocyte migration from adjacent epidermal layers [15,18]. Some authors have hypothesized that coagulation-related genes, such as *antithrombin*, *urokinase*, and *serpine1*, may be involved in the formation of this structure in fishes [16,24]. Additionally, mucous cells were detected alongside migrating keratocytes at the wound borders in the maraena whitefish, confirming earlier observations of mucous cell differentiation during wound re-epithelialization in Atlantic salmon [16]. After re-epithelialization is complete, keratocyte proliferation occurs, leading to the formation and thickening of the neo-epidermis [11,18,25]. During this process, the newly formed neo-epidermis initially contains few mucous cells; however, as keratocytes proliferate, their numbers increase, appearing apically in a bead-on-string arrangement [11,18]. Numerous large mucous cells, open to the extracellular space, were also observed in the examined maraena whitefish on the 3rd dpw, likely indicating the start of an early innate immune defense protecting the neo-epidermis and wound bed against pathogens. As in other fish species, the wounding of maraena whitefish also caused immediate hyperpigmentation with enlarged chromatophores (most likely melanophores) present throughout all time points (especially on the 3rd dpw) only on wounded skin [3]. Considering melanin’s antioxidant properties and its role in protecting against environmental stressors (UV irradiation and oxidizing agents) and pathogens (especially through the synergistic antibacterial and antifungal properties of toxic intermediates generated during melanin biosynthesis), this pigmentation may serve as an early protective mechanism for the wound surface [26,27,28].

Inflammatory response during injury healing plays an important role in clearing the wound bed of damaged tissue and extracellular pathogens, as well as initiating the repair process [3]. In fishes, this response involves an initial influx of neutrophils and other leucocytes (B cells and T cells) to the wound site, migrating from the head kidney, followed by the later arrival of macrophages originating from blood-derived monocytes [12,18,24]. Acute inflammation in fishes is generally triggered by the activation of pro-inflammatory effectors such as chemokines and interleukins (especially, *IL-1*, *IL-6*, *IL-8*, *IL-12*, *cox-2*, and *TNF*), which guide leukocytes to the wound area, along with the activation of proteases and other cellular stress responders [3]. In the current study, histological and transcriptomic analyses revealed acute inflammation between approximately the 1st and 8th dpw in the examined maraena whitefish. This inflammatory response was associated with significant upregulation of the *IL-17D*, *CD-4*, *p53*, *HSP-90*, and *MMP-9* genes, with expression peaking on the 3rd dpw, consistent with findings previously reported in Atlantic salmon [17,18]. *Interleukin 17D*, like other cytokines from family *IL-17*, plays a key role in initiating the inflammatory response in fishes [29,30]. Thus, the observed peak in *IL-17D* gene expression across multiple tissues and organs on the 3 dpw likely reflects a whole-organism activation of inflammation in maraena whitefish. *MMP-9* is a representative of proteases secreted by both keratocytes and macrophages. It plays a pivotal role in degrading extracellular matrix components like laminins and fibrillar collagens, while also modulating inflammation by regulating the activity of cytokines and chemokines [12,16,31,32]. In addition, metalloproteinase activity eases the migration of keratocytes during re-epithelialization and promotes the secretion of growth factors from the wound matrix [33,34]. Several studies have shown that both insufficient and excessive metalloproteinase activity can disrupt the inflammatory response in fish, delaying the formation of repair tissue [17,35,36]. In the present study, a steady upregulation of the *MMP-9* gene was observed at all post-wounding time points, suggesting a positive correlation with the progression of wound-healing process in maraena whitefish. *Heat shock protein 90* (*HSP-90*) belongs to a family of chaperone proteins synthesized in response to cellular stress. It plays an important role in the degradation of damaged cells and wound remodeling during the injury-healing process in animals [37,38]. In experimentally wounded maraena whitefish, *HSP-90* gene expression peaked on the 3rd dpw, indicating that the acute inflammatory response was fully initiated on this day. Interestingly, significant upregulation of the *p53* gene in wounded maraena whitefish was recorded only in the liver and head kidney on the mentioned day post-wounding. The *p53* protein plays a crucial role in defense against systemic autoimmunity by controlling the DNA repair process, cell cycle arrest, and apoptosis [39,40]. Thus, its increased expression recorded in the liver and head kidney of examined maraena whitefish may be necessary for the regulation of lymphocytic cell numbers and the removal of erroneous cells during the intensive production phase of inflammation during the wound healing on the 3rd dpw. Histological analysis also revealed that the inflammatory response in wounded maraena whitefish on this day was accompanied by a surge of open mucous cells and increased melanin pigmentation on the wound surface, potentially providing protection against external harmful factors [3].

Granulation tissue formation and wound remodeling are the final phases of wound healing in animals and involve intensive growth of repair tissue from the wound margins, gradually covering and contracting the wound, and replacing damaged area with new tissue [18,24,41]. During this stage of wound healing, the scale regeneration process is also observed, as pre-osteoblasts proliferate from mesenchymal stem cells and deposit a mineralized matrix [12,18,42]. Granulation tissue formation and pronounced wound remodeling were recorded in the examined maraena whitefish between the 3rd and 15th dpw, representing an intermediate timing compared to zebrafish and Atlantic salmon, in which it forms on the 2nd and 14th dpw, respectively [18,24]. In the examined maraena whitefish, the granulation tissue formation was accompanied by an inflammatory response and intense chromatophore infiltration into the fibrotic tissue, which is known to play a key role in initiation of dermal repair process by stimulating fibroblast recruitment, as well as promoting fibroblasts’ proliferation and growth [3]. In fishes, the repair tissue typically consists of connective tissue, fibroblasts, myofibroblasts, inflammatory and immune cells, and blood vessels. The granulation tissue also exhibits hyperpigmentation due to melanocytes infiltration [3], which was similarly observed in the present study. An inflammatory response was detected in experimentally injured maraena whitefish alongside granulation tissue formation. Numerous studies indicate that inflammation plays a crucial role in both fibroblast recruitment and granulation tissue formation in fishes, as it is essential for activating dermal repair through fibroblast proliferation and the release of growth-stimulating signals from inflammatory cells [3]. As remodeling progresses, changes in the type, amount, and organization of collagen also occur in fishes, contributing to wound scarring and improved tensile strength of the tissue [3,43]. Recent research on Atlantic salmon revealed that wound contraction is facilitated by the presence of distinct collagen fibers with three to four rope-like structures in granulation tissue, likely formed by fibroblast migration from the myocommata, which are the major connective tissue compartment in teleost fish muscle tissue [18]. Myofibroblast-like cells were also observed in the developed granulation tissue of the examined maraena whitefish, supporting their potential involvement in the wound-contraction process in fishes [3]. *The transforming growth factor-β* (*TGF-β*) gene was found to be involved in regulating the movement of fibroblasts and keratocytes to the wound site, acting as a growth factor for fibroblasts, keratinocytes, endothelial cells, and smooth muscle cells [44]. Moreover, *TGF-β* regulates wound remodeling and scarring by activating cell proliferation at the lesion site while dampening ongoing inflammation [45,46,47]. In the present study, significant upregulation of both the *TGF-β* and *MMP-9* genes, together with histological signs of near-complete regeneration, was recorded on the 15th dpw. This indicates that molecular and cellular processes associated with wound healing were still ongoing, even though the wounds were macroscopically indistinguishable from undamaged skin in maraena whitefish on that day. Consistent with the recorded results in the current study, significant upregulation of *TGF-β*, along with several other genes involved in collagen synthesis, fibril maturation, and fibroblasts’ growth, has been reported in Atlantic salmon during the late healing phase [18].

The immune system is pivotal in preserving an organism’s physiological balance by detecting alterations from normal conditions and reacting to injuries, infections, and other environmental stressors [47,48,49]. The adaptive immune response is particularly considered an essential factor supporting the wound-healing process in fishes, where activated lymphocytes play a crucial role in initiation inflammation and fibrotic repair [4,50]. In fishes, immune responses encompass two main types: (1) humoral immunity, which involves antibody production by activated B cells in response to antigens; and (2) cellular immunity, wherein activated cytotoxic T cells release cytokines and directly eliminate pathogens [51,52,53,54]. Significant upregulation of the *CD-4* gene, along with *IL-17D* gene, in the injured skin tissue of maraena whitefish was observed in the present study from the 1st dpw and lasted until the last day of experiment, indicating pronounced activation of strong innate and adaptive immune cellular responses, as is typical for fish experiencing skin damage during the early and late wound-healing stages [3]. The *CD-4* gene encodes a protein known as CD4, which is primarily expressed on the surface of CD4+ T cells (also known as helper T cells) that are a type of T lymphocyte [55]. CD4+ T cells play a key role in immune function by aiding other immune cells, like B cells and cytotoxic T cells, which directly kill pathogens and secrete cytokines that further drive the ongoing immune response, inflammation, and tissue remodeling during the wound-healing process [52,54,56,57]. Additionally, CD4+ T cells are known to play an important role as enhancers of the injury-healing process, driving collagen deposition in the wound [55]. The recorded activation of cellular immunity mediated by cytotoxic T cells within wounded tissue is most probably caused by the exposure of maraena whitefish tissues to an unsterile environment due to inflicted mechanical wounding. Beyond wounded tissue, significant upregulation of *CD-4* gene expression was additionally recorded on the 3rd and 8th dpw in the head kidney of the examined species and may relate to a high rate of monocytopoiesis in this organ, as usually occurs in teleost fishes during the activation of immune response [58]. The recorded slight downregulation of the *CD-4* gene in the damaged skin on the 8th dpw might signify a transitional point between the innate and adaptive immune responses in the examined species.

One important limitation of this study concerns the absence of anesthesia during the skin-puncture procedure. This approach was chosen due to the lack of validated information on the transcriptomic effects of anesthetic agents in maraena whitefish, as the use of such compounds could introduce additional confounding factors into gene expression analyses. Nonetheless, we acknowledge that mechanical wounding without anesthesia may have caused stress or pain, potentially influencing the regulation of immune- and stress-related genes. Although sharp, sterile needles and rapid handling were used to minimize distress, the physiological effects of nociception cannot be fully excluded. Future studies should evaluate the impact of commonly used anesthetics on transcriptomic profiles in this species to improve experimental protocols while aligning with both scientific and welfare standards.

## 4. Materials and Methods

### 4.1. Ethics

This study was carried out in strict accordance with the recommendations in the Polish ACT of 15 January 2015 on the Protection of Animals Used for Scientific or Educational Purposes (Journal of Laws 2015, item 266). The experimental protocol used in this study was approved by the Local Ethical Committee for Experiments on Animals in Olsztyn, Poland (Permission no. 31/2013).

### 4.2. Fish Origin

The maraena whitefish specimens examined in this study were obtained from the Department of Sturgeon Fish Breeding, National Inland Fisheries Research Institute in Olsztyn, located in Pieczarki, Poland (54°6′49.01″ N 21°47′50.07″ E). To investigate the healing process of full-thickness skin wounds, fifty maraena whitefish specimens (*n* = 50) within the first year of life (mean body weight of 3.80 ± 0.27 g and length 8.10 ± 0.34 cm per individual) were used. This experiment was conducted at the Center of Aquaculture and Ecological Engineering, University of Warmia and Mazury in Olsztyn, Poland. For this purpose, the fish were placed in 100 L tanks with a water flow rate of 10 L per minute, operating within a recirculating aquaculture system (RAS) and maintained for a six-week acclimation period under controlled culture conditions. The average water temperature was 11.3 ± 0.2 °C, and the oxygenation level consistently exceeded 80%. The fish were fed Aller Safir 1mm-XS-S (Aller Aqua, Poland) three to four times per day, at a rate of 1.0% of fish biomass.

### 4.3. Experimental Setup and Sampling

Following the acclimation period and health assessment, all fish were randomly stocked into five separate 50 L tanks (ten fish per each). The fish in four tanks were experimentally injured with a dorsal skin puncture just below the dorsal fin (experimental groups), while the fish from the fifth tank remained untreated (control group). A 15G hypodermic needle (outer diameter: ~1.8 mm) was used to create a full-thickness wound penetrating all skin layers, simulating the PIT tagging procedure commonly used in salmonid fish aquaculture production. The needle was inserted perpendicularly to a depth of approximately 2 mm, reaching the underlying musculature. Although direct wound measurements were not taken, all punctures were performed by the same trained researcher at a consistent angle and depth to ensure standardization. A new sterile needle was used for each fish to prevent potential pathogen transmission. After puncture, each fish was photographed and returned to its tank for up to 15 days post-wounding (dpw). Anesthetic agents were not used due to the lack of validated information on their transcriptomic effects in maraena whitefish. This decision was made to avoid introducing unknown variables that might interfere with the gene expression analysis. Nevertheless, we recognize the potential welfare concerns and minimized harm by using sharp sterile needles, ensuring rapid handling and limiting fish exposure time out of water to under 30 s.

For gene expression analysis, five fish were randomly sampled from each of the four different experimental group tanks at every time point (1st, 3rd, 8th, and 15th dpw), and from the control group tank (non-injured fish), resulting in five biological replicates per time point/group (*n* = 25 fish in total). This approach ensured that each tank was sampled only once, minimizing the handling stress on the remaining fish. From each fish, liver, head kidney, and both wounded and intact skin samples were collected and processed individually, without pooling. All the selected fish were humanely sacrificed by cutting the spiral cord before tissue sampling for transcriptomic analysis. Total RNA was extracted separately from each tissue sample, and gene expression was measured using SYBR Green-based qPCR with three technical replicates per reaction. Non-injured fish from the control tank/group were sampled in the same manner (*n* = 5) at the beginning of the experiment (the day after acclimation). Wounded skin was also dissected from the remaining five fish in each test tanks for histological analysis and fixed in Bouin’s solution on the same days as the transcriptomic sampling (*n* = 25 fish in total). Prior to tissue collection for histology, the fish were euthanized with an overdose of MS-222 (500 mg/L; Sigma, Brøndby, Denmark).

### 4.4. RNA Extraction, cDNA Synthesis, and Real-Time PCR Analysis

Total RNA was extracted from tissues preserved in RNAlater using TriPure Isolation Reagent (Roche, Mannheim, Germany), following the manufacturer’s instructions. Tissue homogenization was performed by MagNA Lyser (Roche, Mannheim, Germany). Residual DNA in the extracted RNA samples was removed using recombinant DNase I enzyme (A&A Biotechnology, Gdynia, Poland). RNA concentration and purity were measured using a NanoDrop 2000 photometer (Thermo Scientific, Wilmington, DE, USA), and RNA integrity was assessed by 1% agarose gel electrophoresis. The resulting RNA samples were immediately processed for downstream transcriptomic analysis.

The cDNA synthesis was carried out using the Transcriptor First Strand cDNA Synthesis Kit (Roche, Mannheim, Germany) with isolated RNA templates of confirmed quality. Reaction mixtures were prepared in a total volume of 20 µL, consisting of 1x reaction buffer (8 mM MgCl_2_), 2.5 µM anchored-oligo(dT)_18_ primers, 1 mM of dNTP mix, 20U RNA-se Inhibitor, 10U Reverse Transcriptase, and 1 µg of total RNA sample. Reverse transcription was carried out using a Veriti 96-Well Thermal Cycler (Applied Biosystems, Foster City, CA, USA). Samples were incubated at 55 °C for 30 min, followed by enzyme inactivation at 85 °C for 5 min. The resulting cDNA samples were diluted 1:10 in DEPC-treated water and used for subsequent qPCR analysis.

Real-time PCR analysis was performed using previously designed primers targeting six genes involved in immune response and inflammation (*interleukin 17D* and *cluster of differentiation 4*), cellular stress response and protein folding (*heat shock protein 90*), and cell proliferation and tissue repair (*metalloproteinase-9*, *p53 protein*, and *transforming growth factor-β*) (Table 1). In turn, the *eukaryotic translation initiation factor 3 subunit 6* (*eIF3S6*) was chosen as the housekeeping gene due to its stable expression level across different tissues, regardless of tissue damage and other external environmental conditions’ variation [32,59]. Real-time PCR reactions for housekeeping and target genes were run separately on a LightCycler 480 II system (Roche, Mannheim, Germany) using SYBR Green I Master Mix (Roche, Mannheim, Germany). Each reaction was carried out in a total volume of 10 µL and contained 1× SYBR Green I Master Mix, 150–500 nM of each primer, and 1 µL of diluted cDNA. All real-Time PCRs were run in triplicates under the following thermal cycling conditions: initial polymerase activation at 95 °C for 5 min, followed by 35 cycles of 95 °C for 15 s (denaturation), 56 °C for 10 s (primer annealing), and 72 °C for 15 s (extension). Reaction efficiencies were estimated from the slopes of the standard curves made of 5-point and 10-fold serial cDNA dilutions starting at 50 ng/µL. All primer pairs exhibited amplification efficiencies between 90% and 110%. Negative controls with nuclease-free water and non-reverse-transcribed RNA were included in each run to monitor contamination and potential genomic DNA presence. A melting curve analysis (60–95 °C) was performed at the end of each run to verify amplification specificity. Fluorescence was measured after each elongation step and at 0.5 °C increments during a melting curve analysis. Relative gene expression levels were calculated using the 2^−ΔΔCt^ method based on the difference in Ct values between the target and reference genes, as described by Pfaffl [60].

### 4.5. Histological Observations

Fixed fragments of wounded skin tissue samples in Bouin’s solution were dehydrated using a standard protocol involving sequential transfer through ethanol solutions of increasing concentrations at 35 °C (80%, 2 × 1 h; 95%, 2 × 1 h), followed by 100% ethanol (4 × 1 h). The dehydrated tissues were then placed in xylene (2 × 1.5 h, at 35 °C) and subsequently embedded in paraffin (3 × 1 h, 58 °C, three times) using Leica TP 1020 tissue processor (Leica Biosystems, IL, USA). The paraffin-embedded tissues were sectioned at 4–6 µm thickness using a Leica RM2165 rotary microtome (Leica Microsystems, Wetzlar, Germany). The sections were stained with routine hematoxylin and eosin (H&E) staining [61]. Histological slides were examined under an Axio Scope A1 optical microscope equipped with AxioVs40 v4.8 image analysis software (Carl Zeiss Microscopy GmbH, Jena, Germany). Cellular structures and cell-type nomenclature were adapted from Sveen et al. [3].

### 4.6. Statistical Analysis

The relative quantification values of the examined target genes across sampled tissues were analyzed using Statistica software v.10.0 (StatSoft Inc., Tulsa, OK, USA). Prior to analysis, the normality of data distribution and homogeneity of variances were assessed using the Shapiro–Wilk test and Levene’s test, respectively. A two-way ANOVA was performed, with time (0, 1, 3, 8, and 15 dpw) and tissue (liver, head kidney, wounded skin, and intact skin) as fixed factors. Tukey’s HSD post hoc test was used to determine significant differences (*p* < 0.05) between tissues and time points (groups).

## 5. Conclusions

The present study provides preliminary insights into the skin-healing process in maraena whitefish, a species of significant ecological and aquacultural importance in Europe. The combined histological and gene expression analyses revealed that full-thickness skin wounding in the examined species not only triggers a local dermal response but also induces systemic effects across multiple organs and tissues. Complete skin regeneration after puncture with a 15G hypodermic needle in the examined species required at least 15 days. The healing process in the maraena whitefish consisted of four main stages: (1) re-epithelialization, occurring up to approximately 1st day post-wounding (dpw); (2) inflammation from the 1st to the 8th dpw; (3) granulation tissue formation between about the 3rd and 8th dpw; and (4) wound remodeling, taking place between the 3rd and 15th dpw (Figure 3). The recorded results parallel with the major dermal wound-healing stages observed in other fish species, with the overall healing dynamics being intermediate between those of warm- and cold-water species. However, unlike other salmonids, maraena whitefish showed no prolonged remodeling wound phase and a substantial overlap between the early and late stages of skin-injury healing. It should also be mentioned that the observed wound-healing dynamics and total recovery time in maraena whitefish may also be attributed to the nature of the inflicted wounds, as the technique used did not result in severe tissue loss in this study. Overall, the recorded findings deepen our understanding of epidermal barrier restoration in maraena whitefish and provide a foundation for optimizing welfare-oriented husbandry practices, such as stocking densities, wound-care regimens, or immunomodulatory treatments to enhance the skin integrity and disease resistance of the species in aquaculture. Future studies should explore how factors such as temperature, stocking density, and wound severity affect the healing process in the species to support further improvements in its aquaculture production.

## Figures and Tables

**Figure 1 ijms-26-08315-f001:**
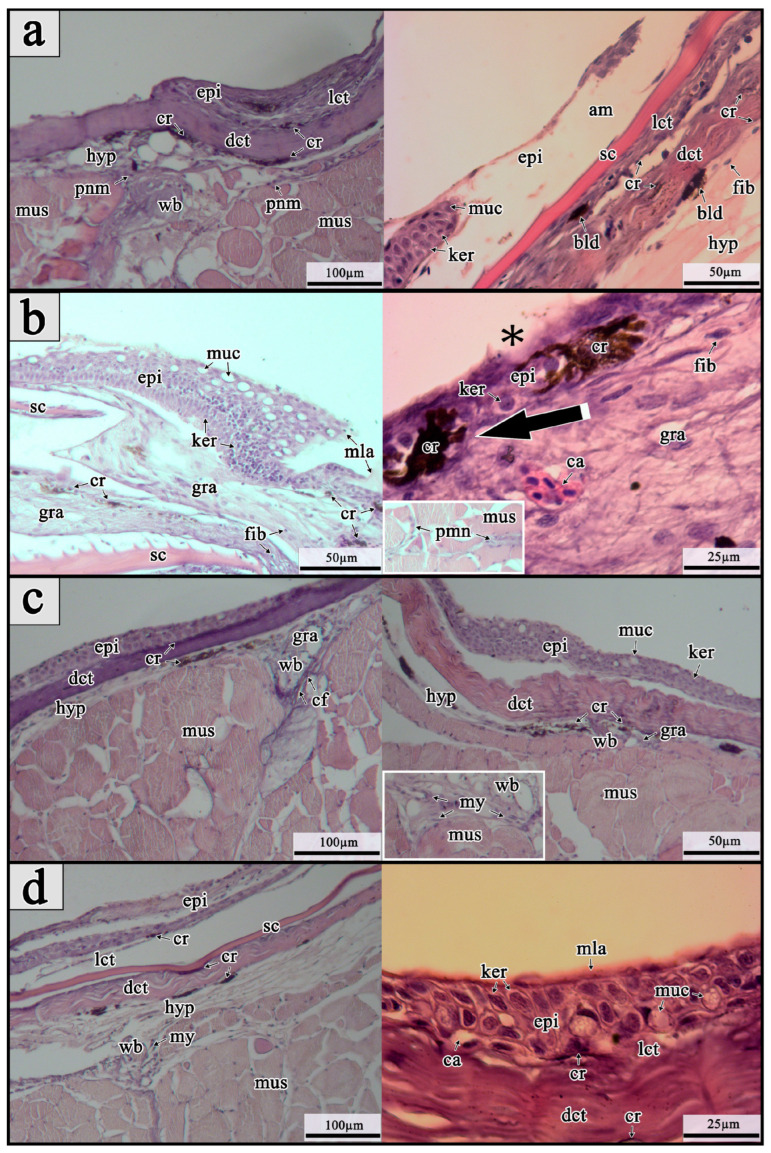
Hematoxylin-and-eosin-stained histological cross-sections of skin tissues showing the wound-healing process on the (**a**) 1st, (**b**) 3rd, (**c**) 8th, and (**d**) 15th days post-wounding (dpw) in maraena whitefish (*Coregonus maraena*) following full-thickness skin injury induced by a 15 G hypodermic needle. Abbreviations: am—amorphous exudate; bld—bleeding; cf—collagen fibrils; cr—chromatophores (probably melanophores); dct—dense connective tissue; epi—epidermis; fib—fibroblasts; gra—granulation tissue; hyp—hypodermis; ker—keratocytes; lct—loose connective tissue; mla—mucus layer; muc—mucous cells; mus—muscles; my—myofibroblast-like cells; pmn—polymorphonuclear inflammatory cells (presumably leukocytes); sc—scale; wb—wound bed. Large black arrow indicates enlarged melanophores, and an asterisk marks the presence of open mucus cells at the epidermis surface.

**Figure 2 ijms-26-08315-f002:**
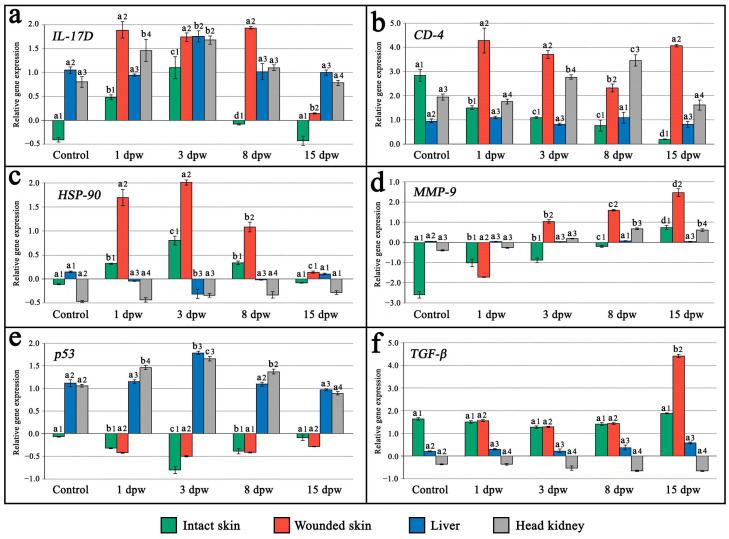
Relative expression levels of (**a**) *interleukin 17D* (*IL-17D*), (**b**) *cluster of differentiation 4* (*CD-4*), (**c**) *heat shock protein 90* (*HSP90*), (**d**) *metalloproteinase-9* (*MMP-9*), (**e**) *p53 protein* (*p53*), and (**f**) *transforming growth factor-β* (*TGF-β*) genes in wounded and intact skin tissues, as well as liver and head kidney samples, collected from experimentally wounded and control maraena whitefish (Coregonus maraena). Different superscript letters (a–d) indicate statistically significant (*p* < 0.05) differences in gene expression levels between sampling times (experimental groups) within the same or organ, while different superscript numbers (1–4) indicate significant differences (*p* < 0.05) between tissues/organs at the same sampling time. Values are presented as fold changes relative to the reference gene (*eukaryotic translation initiation factor 3 subunit 6*, *eIF3S6*). The measure of variation is derived from the respective SEM of the Ct values. Data are shown as the mean with standard deviation.

**Figure 3 ijms-26-08315-f003:**
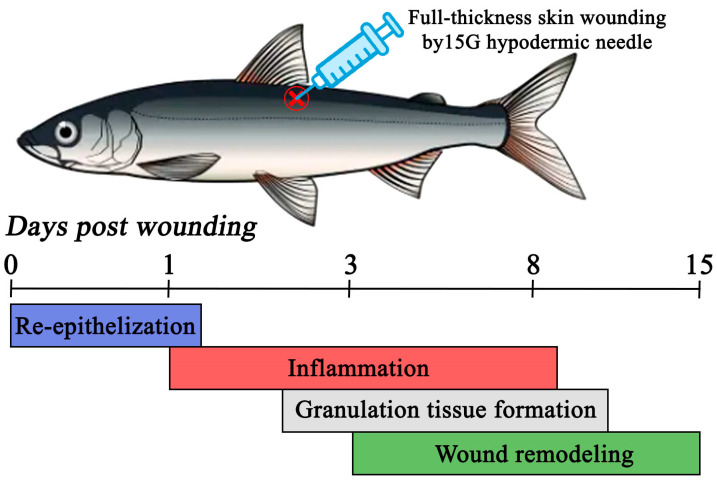
Summary of the wound-healing cascade recorded under the current research in the maraena whitefish (*Coregonus maraena*) subjected to full-thickness skin injury by a 15 G hypodermic needle.

**Table 1 ijms-26-08315-t001:** Forward (F) and reverse (R) primer sequences used for the real-time PCR analysis of the selected genes in the present study.

Gene	Primer Sequences	References
*Interleukin 17D* (*IL-17D*)	F: TGGGCCTACAGGCTGAATTA R: GACCAGACAGCCCTTACAAA	[30]
*Cluster of differentiation 4* (*CD-4*)	F: TGCATTGTTCCTCTCTTCCACAGC R: CCGTCCCAAGGTACCATAGTACCAA	[32]
*Heat shock protein 90* (*HSP-90*)	F: GAACCTCTGCAAGCTCATGAAGGA R: ACCAGCCTGTTTGACACAGTCACCT	[32]
*Metalloproteinase-9* (*MMP-9*)	F: AGTCTACGGTAGCAGCAATGAAGGC R: CGTCAAAGGTCTGGTAGGAGCGTAT	[32]
*p53 protein* (*p53*)	F: CGAGCCCTGGCCGTCTATAA R: GGGCAGGACCTTCATTGTTC	[39]
*Transforming growth factor-β* *(TGF-β*)	F: AATCGGAGAGTTGCTGTGTGCGA R: GGGTTGTGGTGCTTATACAGAGCCA	[32]
*Eukaryotic translation initiation factor 3 subunit 6* (*eIF3S6*)	F: GTCGCCGTACCAGCAGGTGATT R: CGTGGGCCATCTTCTTCTCGA	[32]

## Data Availability

The raw data supporting the conclusions of this article will be made available by the authors upon request.

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
