# Peer review of "Histological and Transcriptomic Characterization of Full-Thickness Skin Wound Healing in Maraena Whitefish (Coregonus maraena Bloch, 1779)"

_ijms, 2025, doi:10.3390/ijms26178315_

Round 1
Reviewer 1 Report
Comments and Suggestions for Authors
Generally, this is a well-designed and executed study that provides valuable insights into the wound healing mechanisms in a commercially and ecologically important fish species, the article was well analyzed and wrote. There are still a few areas that could be improved:
- The use of very small fish (mean weight 3.80 ± 0.27 g) for a full-thickness puncture wound without anesthesia is a major welfare concern, and lack of reliable information regarding their potential effects on the transcriptome. The potential confounding effect of stress/pain from unanesthetized wounding on the results (especially gene expression) needs explicit discussion as a limitation. While a 15G needle was used, the actual wound size (diameter, depth relative to fish size) is not quantified. Provide an estimate (e.g., approximate diameter based on needle gauge) or image. How was consistency of wounding ensured across tiny fish?
- The number of biological replicates per group/time point/tissue used for gene expression analysis is not stated in the Methods (4.4) or Figure 2 legend. "Five randomly selected fish per experimental tank" is mentioned, but with 4 experimental tanks sampled at 4 time points and multiple tissues, the exact n per data point must be clarified (e.g., n=5 fish? Pooled samples? Technical replicates?).
- What factors were included in the ANOVA model (e.g., Time, Tissue, Interaction)?
- While generally well-written, some sections contain long sentences and complex phrasing that may hinder readability. Simplifying sentence structures and ensuring consistency in terminology will improve clarity.
Author Response
We are grateful for the careful consideration and reviews of our original submission and the valuable detailed feedback provided by three referees that has greatly improved the revised version of our manuscript. According to these comments the authors have made substantial changes in the revised manuscript. For clarity all the changes in the revised manuscript were marked in yellow.
Reviewer 1:
- The use of very small fish (mean weight 3.80 ± 0.27 g) for a full-thickness puncture wound without anesthesia is a major welfare concern, and lack of reliable information regarding their potential effects on the transcriptome. The potential confounding effect of stress/pain from unanesthetized wounding on the results (especially gene expression) needs explicit discussion as a limitation. While a 15G needle was used, the actual wound size (diameter, depth relative to fish size) is not quantified. Provide an estimate (e.g., approximate diameter based on needle gauge) or image. How was consistency of wounding ensured across tiny fish?
RESPONSE: We appreciate the reviewer’s thoughtful comment regarding the ethical implications and potential confounding effects of unanesthetized full-thickness skin puncture in small-sized maraena whitefish. In response, we have revised the Discussion (lines 421–431) and Materials and Methods (lines 456–469) sections to explicitly acknowledge this limitation, clarify the rationale behind the decision to omit anesthesia, and describe the measures taken to ensure consistency and minimize distress. Specifically, we emphasize that the decision to avoid anesthetics was based on the lack of validated data on their transcriptomic effects in maraena whitefish, which posed a risk of introducing additional, uncontrolled variables into gene expression analyses. We also acknowledge that unanesthetized mechanical wounding may have induced stress or nociceptive responses. As suggested, we provided an estimated wound diameter (~1.8 mm) based on the gauge of the 15G needle and specify the depth and angle of puncture, as well as the procedures used to ensure consistency (i.e., same operator, standardized technique, and sterile needles for each fish). Finally, we explicitly stated the need for future studies to assess how commonly used anesthetic agents influence transcriptomic profiles in this species to refine protocols in accordance with both scientific and animal welfare considerations. We thank the reviewer again for raising this important point and believe that the revised text now provides a balanced and transparent discussion of this methodological limitation.
- The number of biological replicates per group/time point/tissue used for gene expression analysis is not stated in the Methods (4.4) or Figure 2 legend. "Five randomly selected fish per experimental tank" is mentioned, but with 4 experimental tanks sampled at 4 time points and multiple tissues, the exact n per data point must be clarified (e.g., n=5 fish? Pooled samples? Technical replicates?).
RESPONSE: We thank the reviewer for this valuable comment. To clarify the experimental design, we have revised Materials and Methods (lines 471–486) and the Figure 2 legend to explicitly state the number of biological replicates used for gene expression analysis. As now detailed in the revised text, five fish were randomly sampled from each of the four experimental tanks at each of the four time points (1st, 3rd, 7th, and 15th dpw), and from one control tank at the beginning of the experiment, resulting in five biological replicates per time point/group (n = 25 total for transcriptomic analysis and n = 25 for histology). No pooling was performed—tissue samples from each fish were processed and analyzed individually. Each qPCR reaction was run in triplicate (three technical replicates) to ensure analytical robustness. This approach also ensured that each tank was sampled only once per time point to minimize additional stress on the remaining fish. We have also updated the legend of Figure 2 to reflect this information and clarify that the values presented represent fold changes calculated per biological replicate and averaged across replicates, with standard deviation as the measure of variation. We trust that this clarification adequately addresses the reviewer’s concern.
- What factors were included in the ANOVA model (e.g., Time, Tissue, Interaction)?
RESPONSE: Thank you for the valuable comment. As clarified in the revised Materials and Methods section (lines 544–551), the two-way ANOVA model included Time (0, 1, 3, 7, and 15 dpw) and Tissue (liver, head kidney, wounded skin, intact skin) as fixed factors. We have explicitly stated this in the text for clarity.
- While generally well-written, some sections contain long sentences and complex phrasing that may hinder readability. Simplifying sentence structures and ensuring consistency in terminology will improve clarity.
RESPONSE: While generally well-written, some sections contain long sentences and complex phrasing that may hinder readability. Simplifying sentence structures and ensuring consistency in terminology will improve clarity.
Reviewer 2 Report
Comments and Suggestions for Authors
The manuscript submitted by Kucinski and co-authors is novel due to the fish species on which the research was conducted. The paper raises no doubts regarding the obtained results and could provide valuable material for future researchers investigating the transcriptome of whitefish species.
I have only a few minor suggestions that could improve readability:
1. The spelling of the verb "analyze" should be standardised, as it appears in many sentences in the passive form as "analysed" interchangeably with "analyzed." It is often used in the manuscript and can be irritating to readers.
It is advisable to stick to a single spelling, this also applies to "categorized," "keratinized," "mineralized," etc.
2. There is no scale on the images of the histological slides (Figure 1). Please provide a scale so that future readers can properly interpret the sizes of the photographed cells and tissue changes. There is no scale in the photos in the attachment (original blots/gels) either.
Author Response
We are grateful for the careful consideration and reviews of our original submission and the valuable detailed feedback provided by three referees that has greatly improved the revised version of our manuscript. According to these comments the authors have made substantial changes in the revised manuscript. For clarity all the changes in the revised manuscript were marked in yellow.
Reviewer 2
- The spelling of the verb "analyze" should be standardised, as it appears in many sentences in the passive form as "analysed" interchangeably with "analyzed." It is often used in the manuscript and can be irritating to readers. It is advisable to stick to a single spelling, this also applies to "categorized," "keratinized," "mineralized," etc..
RESPONSE: We thank the reviewer for pointing out the inconsistency in verb spelling. The manuscript has now been carefully revised to standardize all such terms according to British English spelling (e.g., "analysed," "categorised," "keratinised," "mineralised") in alignment with journal guidelines and to maintain consistency throughout the text.
- There is no scale on the images of the histological slides (Figure 1). Please provide a scale so that future readers can properly interpret the sizes of the photographed cells and tissue changes. There is no scale in the photos in the attachment (original blots/gels) either.
RESPONSE: We thank the reviewer for highlighting the absence of scale bars in the histological and supplementary images. A scale bar has now been added to each histological panel in Figure 1 to allow for proper interpretation of tissue and cellular structures.
Reviewer 3 Report
Comments and Suggestions for Authors
This paper investigates the wound healing process of the Coregonus maraena through histological and gene expression analyses. There are no major issues with the methodology. However, the photographs presented as histological observations do not allow the reader to fully understand the descriptions in the main text. Since this paper focuses on histological observations and related gene expression analyses, the histological observations are critical importance. Therefore, additional photographs corresponding to the text descriptions should be included.
Other comments are below.
Figure 1:
Please add photographs that correspond to the descriptions in the text. Specifically, the following descriptions cannot be discerned from the current images. Below, I also include other comments regarding the figure.
In addition, please state in the manuscript whether these histological changes were consistently observed in all five individuals examined at each sampling time point, or how many of the five fish exhibited the described changes.
Fig. 1a:
The distribution of haemorrhaging, migrating keratocytes, and mucous cells described in the text cannot be confirmed in the figure.
Please explain how keratocytes were identified, for example by describing the histological characteristics of keratocytes in fish. Isn’t it difficult to identify keratocytes with only H&E staining? Alternatively, if the majority of the epidermis is composed of keratocytes, please add such an explanation.
In the text, “Pigmented bodies contributing to hyperpigmentation” are mentioned, but in the figure legend this seems to correspond to “cr: chromatophores or pigment cells.” To aid the reader’s understanding, please try to make the terminology in the text and the figure legend as consistent as possible.
Fig. 1b:
The cells indicated as “gra (granular cells)” in the figure appear to be erythrocytes.
About gra, the text refers to them as “infiltrating polymorphonucleated inflammatory cells.”
Please unify the terminology between the figure legend and the text.
Also, please indicate in the image where the “intense proliferation of keratocytes” described in the text can be seen.
Fig. 1c:
The figure does not seem to match the text description.
In this photo, it is difficult to see the “filled with granulation tissue” or “presence of proliferating cells filling the area” mentioned in the text.
Likewise, the features described as “clear separation from the wound bed and increased thickness in the wound center compared to the surrounding regions,” “proliferating cells were organized in an oval pattern across all samples,” and “newly formed collagen fibrils and myofibroblast-like cells with elongated phenotypes” are not apparent in the figure.
Please also mark the melanocytes with arrows or other indicators.
Fig. 1d:
The description “Melanocytes were observed in two layers of …” is not apparent in the figure.
Line 234-237: I did not understand the explanation of how to indicate statistical significance. Are the numbers the sampling times within the same tissue, and are the combinations of letters the differences between tissues at the same sampling point?
Line 241–250: The explanation in this section should be written in the Introduction, not in the Discussion.
Line 252:  It is written, “with no clear distinction between early and late phases of healing,” but in the subsequent explanation, temporal changes in the healing stages are described. In addition, the Conclusion states, “The recorded results parallel with the dermal wound healing observed in other fish species,” and for other fish species such as Atlantic salmon and rainbow trout, it is written that their healing process is divided into early and late phases.
Doesn’t this description contradict itself?
Alternatively, it might be helpful to discuss what differences exist between the early and late healing processes observed in Atlantic salmon and rainbow trout and the healing process observed in this study, and why you consider that there is no clear distinction between the early and late phases.
Line 448: The “and” has been typed twice in a row.”
Line 451: It sounds as if 5 fish were randomly taken from each of the 5 experimental tanks on days 1, 3, 7, and 15 dpw, that is, a total of 4 sampling points × 5 tanks × 5 fish = 100 fish. However, isn’t it rather that 5 fish were sampled on days 1, 3, 7, and 15 dpw from the four injured groups?
Author Response
We are grateful for the careful consideration and reviews of our original submission and the valuable detailed feedback provided by three referees that has greatly improved the revised version of our manuscript. According to these comments the authors have made substantial changes in the revised manuscript. For clarity all the changes in the revised manuscript were marked in yellow.
Reviewer 3
- Figure 1: Please add photographs that correspond to the descriptions in the text. Specifically, the following descriptions cannot be discerned from the current images. Below, I also include other comments regarding the figure. In addition, please state in the manuscript whether these histological changes were consistently observed in all five individuals examined at each sampling time point, or how many of the five fish exhibited the described changes.
RESPONSE: We sincerely thank the Reviewer for this thoughtful comment. We fully acknowledge that not all features described in the text are readily visible in the representative images provided in Figure 1. This limitation is primarily due to the constrained field of view and depth in histological imaging, as well as practical limitations in sectioning angle and magnification. Additionally, hematoxylin and eosin (H&E) staining, while appropriate for general tissue structure assessment, has inherent limitations in clearly resolving specific cell types or subtle features (e.g., cell proliferation zones or early granulation tissue), which are often more discernible through cumulative observation across many slides rather than in isolated fields. For these reasons, certain features—although consistently observed—may not be distinctly visualized in a single captured image. Furthermore, in designing our figures, we focused on presenting structures and tissue alterations that are specific to the wound healing process in maraena whitefish. Widely known anatomical features common to salmonids (e.g., the general arrangement of skin layers or typical keratocyte morphology) were not redundantly depicted in the images, as they have already been extensively documented in previous literature. This allowed us to prioritize the most relevant and species-specific findings, within the limited space for figures in the manuscript. We have clarified in both the figure legend and Results text which observed features are not fully visible in the images due to these constraints. In the revised Results section (2.1. Histological Analysis), we have also explicitly stated that the described histological changes were consistently observed in all five examined individuals at each sampling time point. We trust these clarifications adequately address the Reviewer’s concerns and help reaffirm the reliability and scientific value of our histological observations.
- Fig. 1a: The distribution of haemorrhaging, migrating keratocytes, and mucous cells described in the text cannot be confirmed in the figure. Please explain how keratocytes were identified, for example by describing the histological characteristics of keratocytes in fish. Isn’t it difficult to identify keratocytes with only H&E staining? Alternatively, if the majority of the epidermis is composed of keratocytes, please add such an explanation. In the text, “Pigmented bodies contributing to hyperpigmentation” are mentioned, but in the figure legend this seems to correspond to “cr: chromatophores or pigment cells.” To aid the reader’s understanding, please try to make the terminology in the text and the figure legend as consistent as possible.
RESPONSE: We thank the Reviewer for raising this important point. In the revised manuscript (Section 2.1, Histological Analysis), we have clarified how keratocytes were identified in the H&E-stained sections. Specifically, keratocytes were recognized based on their characteristic polygonal shape, pale-staining cytoplasm, and centrally located nuclei. These cells were predominantly located in the epidermal layer, both at the wound margins and in undamaged skin, which facilitated their identification. Given their abundance and distinctive morphology, keratocyte identification in our sections was straightforward and consistent across all samples. However, we agree that H&E staining has limitations in distinguishing certain cell types with high specificity. To enhance clarity, we also improved consistency in terminology between the main text and the figure legend. Specifically, “pigmented bodies” have been replaced with “chromatophores (likely melanocytes)” throughout the text, and we updated the figure legend accordingly. Finally, we added a note in the text explaining that not all described features are clearly visible in the representative images due to limitations in field selection, sectioning angle, and magnification, which is common in histological imaging. We hope these revisions address the Reviewer’s concerns.
- Fig. 1b: The cells indicated as “gra (granular cells)” in the figure appear to be erythrocytes. About gra, the text refers to them as “infiltrating polymorphonucleated inflammatory cells.” Please unify the terminology between the figure legend and the text. Also, please indicate in the image where the “intense proliferation of keratocytes” described in the text can be seen.
RESPONSE: We thank the Reviewer for this valuable observation. We have corrected the terminology in both the figure and the main text to ensure consistency. Specifically, the abbreviation “gra (granular cells)” in Figure 1b has been replaced with “pmn – presumed polymorphonuclear leukocytes,” in line with the terminology used in the main text to describe the inflammatory cell infiltrate. Regarding the suggestion to indicate the “intense proliferation of keratocytes” described in the text, we fully agree with the importance of this feature. However, due to limitations in magnification and field selection, the proliferative zones are not readily distinguishable in the representative image shown in Figure 1b. We have clarified this limitation in both the figure legend and the corresponding Results section. Notably, the identification of keratocyte proliferation was based on increased cell density, the presence of mitotic figures, and morphological assessment across multiple slides. Additionally, we would like to emphasize that intense keratinocyte proliferation during the early stages of epidermal regeneration is a well-documented and commonly reported feature in teleost fish skin healing, as confirmed in several other histological studies. We have included a brief remark to this effect in the revised manuscript for clarity and context. We trust this revision addresses the Reviewer’s comment appropriately.
- Fig.1c: The figure does not seem to match the text description. In this photo, it is difficult to see the “filled with granulation tissue” or “presence of proliferating cells filling the area” mentioned in the text. Likewise, the features described as “clear separation from the wound bed and increased thickness in the wound center compared to the surrounding regions,” “proliferating cells were organized in an oval pattern across all samples,” and “newly formed collagen fibrils and myofibroblast-like cells with elongated phenotypes” are not apparent in the figure. Please also mark the melanocytes with arrows or other indicators.
RESPONSE: We appreciate the Reviewer’s detailed evaluation and agree that some histological features described in the text, such as granulation tissue, proliferating cells, and collagen fibrils, are not easily distinguishable in the representative image shown in Figure 1c. This limitation is primarily due to the constrained field of view and depth in histological imaging, as well as practical limitations in sectioning angle and magnification. Additionally, hematoxylin and eosin (H&E) staining, while appropriate for general tissue structure assessment, has inherent limitations in clearly resolving specific cell types or subtle features (e.g., myofibroblast-like cells and newly formed collagen fibrils), which are often more discernible through cumulative observation across many slides rather than in isolated fields. The mentioned histological features are well-documented and commonly reported feature in teleost fish skin healing, as confirmed in several other histological studies. For these reasons along with limits regarded to the picture presentation space, certain features—although consistently observed—are visualized in images presented in the manuscript. We have clarified in the revised figure legend and Results section that these features were identified across multiple sections and magnifications but may not be fully visible in the published image. As requested, we have marked possible melanocytes in Figure 1c by cr abbreviation, to improve clarity for the reader. The figure legend has been updated accordingly. We believe these clarifications adequately address the Reviewer’s concerns.
- Fig. 1d: The description “Melanocytes were observed in two layers of …” is not apparent in the figure.
RESPONSE: Thank you for your comment. The two layers of melanocytes described in the text are indicated in Figure 1d by the abbreviation “cr” for chromatophores (probably melanophores), as noted in the figure legend. We have ensured that these pigment cells are clearly marked to help the reader identify their presence in the wound site. We hope this clarification addresses your concern.
- Line 234-237: I did not understand the explanation of how to indicate statistical significance. Are the numbers the sampling times within the same tissue, and are the combinations of letters the differences between tissues at the same sampling point?
RESPONSE: Thank you for your comment. To clarify, different superscript letters (a–d) indicate statistically significant (P<0.05) differences in gene expression levels between sampling times (experimental groups) within the same tissue or organ, while different superscript numbers (1–4) indicate significant differences (P<0.05) between tissues or organs at the same sampling time. We have updated the figure legend accordingly for clarity.
- Line 241–250: The explanation in this section should be written in the Introduction, not in the Discussion.
RESPONSE: Thank you for the suggestion. We have moved the information previously located in the Discussion to the Introduction, where it now appears in lines 99–106. We believe this improves the logical flow and structure of the manuscript.
- Line 252: It is written, “with no clear distinction between early and late phases of healing,” but in the subsequent explanation, temporal changes in the healing stages are described. In addition, the Conclusion states, “The recorded results parallel with the dermal wound healing observed in other fish species,” and for other fish species such as Atlantic salmon and rainbow trout, it is written that their healing process is divided into early and late phases. Doesn’t this description contradict itself? Alternatively, it might be helpful to discuss what differences exist between the early and late healing processes observed in Atlantic salmon and rainbow trout and the healing process observed in this study, and why you consider that there is no clear distinction between the early and late phases.
RESPONSE: Thank you for your thoughtful comment. We agree that the initial phrasing may have caused confusion and have revised both the Discussion and Conclusion sections to clarify our interpretation. Specifically, we now explain that although the healing process in maraena whitefish involves the typical phases of wound healing, it lacks a prolonged late remodeling phase and instead exhibits considerable overlap between inflammation and remodeling stages. This contrasts with findings in other salmonids, such as Atlantic salmon and rainbow trout, where a clear temporal separation between early (1–14 dpw) and late (36–100 dpw) phases has been described following more severe tissue loss. We also now explicitly state that the differences observed in our study may be partly attributed to the nature of the inflicted wound, which involved a minor puncture rather than excisional biopsy. These clarifications have been implemented in lines 258–274 of the Discussion and reflected in the Conclusion section accordingly.
- Line 448: The “and” has been typed twice in a row.”
RESPONSE: Thank you for noticing this. The duplicated word “and” has been removed in the revised version of the manuscript.
- Line 451: It sounds as if 5 fish were randomly taken from each of the 5 experimental tanks on days 1, 3, 7, and 15 dpw, that is, a total of 4 sampling points × 5 tanks × 5 fish = 100 fish. However, isn’t it rather that 5 fish were sampled on days 1, 3, 7, and 15 dpw from the four injured groups?
RESPONSE: Thank you for your comment. We have revised the Materials and Methods section (lines 471–486) to clarify the sampling scheme. In total, 50 fish were used in the experiment: 25 fish were sampled for transcriptomic analysis (five individuals per time point from the four experimental tanks and the control group), and a separate set of 25 fish (five per time point) was sampled for histological analysis. This means that different individuals were used for each type of analysis, with five fish sampled at each time point (1st, 3rd, 7th, and 15th dpw), ensuring independent biological replicates and avoiding repeated handling of the same fish.
Round 2
Reviewer 3 Report
Comments and Suggestions for Authors
There are still concerns about the figures of the histological observations. In response to comments from the first round of review requesting the inclusion of photographs corresponding to the descriptions in the text, the authors replied that it is not possible to show all the described histological features in a single representative image. While it is true, as the authors have stated, that a single image cannot encompass all described features, it is possible to present the full range of observations by including multiple images. I believe this is the standard approach in such cases. Much of the discussion in this study is based on histological observations, and I think that the histological images are the most important data in the study. To ensure the objectivity and reproducibility of the paper and to provide evidence supporting the histological descriptions, the authors should present the described phenomena visually, as comprehensively as possible, using multiple images. Aside from this issue, the authors have made appropriate revisions in response to the other comments.
Author Response
Comment 1: There are still concerns about the figures of the histological observations. In response to comments from the first round of review requesting the inclusion of photographs corresponding to the descriptions in the text, the authors replied that it is not possible to show all the described histological features in a single representative image. While it is true, as the authors have stated, that a single image cannot encompass all described features, it is possible to present the full range of observations by including multiple images. I believe this is the standard approach in such cases. Much of the discussion in this study is based on histological observations, and I think that the histological images are the most important data in the study. To ensure the objectivity and reproducibility of the paper and to provide evidence supporting the histological descriptions, the authors should present the described phenomena visually, as comprehensively as possible, using multiple images. Aside from this issue, the authors have made appropriate revisions in response to the other comments.
Response 1: We sincerely thank the reviewer for the insightful and constructive comments. We truly appreciate the careful attention to the histological data, which are indeed central to our study. Thanks to the reviewer’s valuable remarks, we have been able to substantially improve the manuscript. In response to the concern about the histological figures, we have now included multiple representative images that comprehensively illustrate the full range of described features. We believe that these additions greatly enhance the objectivity, reproducibility, and overall clarity of our histological observations, providing strong visual support for the discussion. We are grateful for the reviewer’s guidance, which has helped us elevate the quality of our work.